# REGULATING THE LEVEL OF MANIPULATION IN TEXT AUGMENTATION WITH SYSTEMATIC ADJUSTMENT AND ADVANCED SENTENCE-EMBEDDING

## ABSTRACT

Text augmentation, a method for generating samples by applying combinations, noise, and other manipulations to a small dataset, is a crucial technique in natural language processing (NLP) research. It introduced diversity into the training process, thereby enabling the construction of robust models. The level of manipulation is the most important issue in text augmentation; low-level manipulation generates data similar to the original, resulting in inefficient augmentation because it cannot ensure diversity, whereas high-level manipulation causes reliability issues for labels and degrades the model's performance. Therefore, this paper proposes a systematically adjustable text augmentation technique to address the "level of manipulation" issue. Specifically, it proposes a method for systematically adjusting the data candidate pool for manipulation to provide an appropriate level of randomness during the augmentation process. Furthermore, we propose an advanced sentence-embedding methodology to achieve robust pseudo-labeling at the manipulation level. In other words, we leverage combined sentence embedding, which incorporates sentence embedding, document embedding, and XAI information from the original data to assign reliable pseudo-labels. We conducted performance comparisons with existing text augmentation approaches to validate the effectiveness of our proposed methodology. The experimental results demonstrate that the proposed method achieves the highest performance improvement across all the experimental datasets.

## 1 INTRODUCTION

Securing sufficient training data is one of the most critical factors for improving model performance. Consequently, there has been active research on text augmentation, where various transformations are applied to input text data to generate additional data and objects to build accurate and robust models. Text augmentation can be broadly classified into three methods: reorganization-based, which involves transformation insertion, deletion, and reordering applied to the input data; embedding-level methods, such as replacement and mix-based embedding replacement; and deep learning-based Translation and Generation methods. In this study, we focus on the most widely used reorganization methodology and propose a novel approach that differs from previous research that primarily concentrates on reorganization within individual documents. Instead, we introduce a reorganization augmentation method across multiple documents.

One of the most critical issues in Text Augmentation is the level of manipulation. In other words, reducing the manipulation level results in a text generation that closely resembles the original text, leading to decreased diversity and inefficient augmentation. Conversely, increasing the level of manipulation introduces significant variations in the meaning or emotions of the original text, making it challenging to effectively utilize the original text labels and raising concerns about the reliability of the newly assigned pseudo-labels.

Therefore, in this study, we propose a systematically adjustable text augmentation methodology to address manipulation issues. Specifically, we considered the semantic similarity between the seed and selected sentences to systematically adjust the range of the candidate sentence pool. Thus, we can assign appropriate randomness and level of manipulation to augment the data.

More specifically, we arranged all sentences in descending order based on their cosine similarity with the seed sentence and then utilized a logistic regression-based equation with the transformed cosine similarity values between the seed sentence and the selected sentences as input to systematically adjust the range of the candidate sentence pool. In other words, when the similarity between the seed sentence and selected sentences is high, it induces a wide candidate sentence pool, leading to higher randomness and facilitating the selection of diverse sentences. By contrast, when the semantic similarity of the selected sentences is low, a limited candidate sentence pool with lower randomness is assigned, restricting the selection to sentences with meanings similar to the seed sentence. Consequently, our approach offers an appropriate level of manipulation, enabling the systematical adjustment of the manipulation level, which is a critical issue in text augmentation. To the best of the authors' knowledge, this is the first study to propose a systematically adjustable text augmentation method.

In addition, this study proposes an advanced text-embedding methodology for pseudo-labeling that is robust to the level of manipulation. Unlike previous studies that used naive interpolation for the pseudo-labeling of augmented data, this study proposes a reliable model even if the manipulation level is high through text embedding, which combines contextual information such as sentence embedding, document embedding, and XAI information of the original data.

## 2 RELATED WORK

### 2.1 TEXT AUGMENTATION-RELATED STUDIES

Several approaches were initially explored in experiments on vocabulary-replacement-based text augmentation. There is a reorganization-based approach in which direct manipulations are applied to the input data to replace significant text at the word or sentence level, and the embedding-space-based approach, which manipulates and mixes text in the embedding space. A generative-based approach involves using neural machine translation models and pretrained language models (PLM) to generate text.

Re-organization-based augmentation has been widely used in various fields since the research conducted by Wei & Zou (2019). Many studies have explored different approaches, including random deletion, replacement, and insertion of words within the same sentence (Xu et al., 2020; Liu et al., 2020; Dai & Adel, 2020; Karimi et al., 2021), as well as the substitution of existing words with semantically similar alternatives (Marivate & Sefara, 2020; Gao et al., 2019). Moreover, embedding space-based augmentation has been extensively researched, encompassing mix-up-based techniques that mix multiple sentences to increase data variation (Wang et al., 2018; Guo et al., 2020; Zhang et al., 2020b), and studies using self-supervised learning and unsupervised learning methods (Ng et al., 2020; Xie et al., 2020; Chen et al., 2020; Kim & Kang, 2022). Finally, generative-based augmentation has seen diverse researched, such as the back-translation-based approach, which translates source language to target language and trains the model using both the original and translated texts (Edunov et al., 2018; Hayashi et al., 2018; Ibrahim et al., 2020; Ding et al., 2021; Sugiyama & Yoshinaga, 2019), and the methodology of utilizing generated data from pre-trained language models as augmentation data (Wu et al., 2019; Atliha & Šešok, 2020; Liu et al., 2020; Yoo et al., 2020; Anaby-Tavor et al., 2020; Radford & Wu, 2019; Zhang et al., 2020a).

One of the most critical issues in text augmentation is the level of manipulation, particularly in the reorganization-based approach. Reducing the manipulation level generates text that closely resembles the original text, resulting in decreased diversity and inefficient augmentation. This limits the potential contribution to improving the model performance. However, increasing the level of manipulation introduces significant variations in the meaning or sentiments of the original text, making it challenging to use original text labels. This can lead to reliability issues with newly assigned pseudo-labels and potentially cause a decline in the model performance.

Wei & Zou (2019); Karimi et al. (2021); Ding et al. (2021) and Sugiyama & Yoshinaga (2019) methods mentioned above perform manipulations at a very low level because they utilize the original label. Therefore, in studies that perform low-level manipulation, text augmentation is inefficient because the augmented and original data must be semantically similar.

Conversely, a high level of manipulation leads to credibility issues with pseudo-labels for augmented data. According to the experiments conducted by Kumar et al. (2020), in transformer-based methods, the labels of the data change by approximately 40%. Most embedding-based studies rely on simple interpolation-based pseudo-labeling, resulting in limited reliability. To address this problem, some studies Kwon & Lee (2023); Yu et al. (2023) considered the importance of manipulated words, whereas others utilized methods involving unlabeled data (Lee et al., 2013; Berthelot et al., 2019; Shim et al., 2020; Yu et al., 2022). However, despite these efforts, the credibility issue with pseudo-labeling persists as naive methodologies such as weighted sum are utilized.

To address the limitations of the aforementioned studies, we propose a text augmentation methodology that applies a systematical assignment of randomness to control the manipulation level. In addition, to assign robust pseudo-labels based on the manipulation level, we introduced an advanced text-embedding methodology.

## 2.2 EXPLAINABLE ARTIFICIAL INTELLIGENCE

Explainable Artificial Intelligence (XAI) aims to establish a relationship between the inputs and outputs of black-box models that are understandable to humans. In the field of deep learning-based explainable models, there are three major approaches: **Activation-Based Method (ABM)**, **Backpropagation-Based Method (BBM)**, and **Perturbation-Based Method (PMB)**. ABM uses activation weights that are linearly combined in each convolutional layer. CAM(Zhou et al., 2016) and Grad-CAM++(Chattopadhay et al., 2018) are the most representative methods in this category. In contrast, the BBM calculates the magnitude of errors through back-propagation with respect to the input of the model and indicates the importance of each pixel. However, the BBM tends to have less accurate interpretability than the ABM because of its fast computational capability. Representative examples of BBM include the LRP(Montavon et al., 2019) and SmoothGrad(Daniel et al., 2017). Finally, PMB analyzes the model results by adding noise to the input. LIME(Ribeiro et al., 2016) and SHAP(Lundberg & Lee, 2017) are well-known examples of PMB.

To generate text embeddings that include XAI information, we leverage the perturbation-based algorithm SHAP, which is commonly used with text data. SHAP (SHapley Addictive exPlanation) is an XAI methodology based on Shapley Value and local interpretation. It enables interpretation across the entire data domain and analyzes the impact on the model's output by making small changes to the input data and provides interpretability. To simplify the definition using Shapley value's conditional mean, computes the conditional mean of black-box model to derive the SHAP value, which represents the influence of each attribute of the model and input data on the output. In this study utilize the SHAP values of each token present in review texts to create sentence-level embedding vectors, which combine sentence embedding, document embedding, and XAI information.

| Symbol | Typical meaning |
|---|---|
| $E_D$ | Document embedding combining Token embedding and SHAP Values |
| $E_S$ | Sentence embedding combining Token embedding and SHAP Values |
| $\hat{E}$ | Concatenate embedding of $E_D$ and $E_S$; Element of Candidate sentences |
| $\hat{D}$ | Stack embedding for $\hat{E}$ in original data |
| $\hat{A}$ | Stack embedding for $\hat{E}$ in augmented data |
| $\alpha$ | Graph slope; first hyperparameter of Range ratio |
| $m$ | x-axis parallel translation; second hyperparameter of Range ratio |
| $\|\cdot\|$ | Norm; Euclidean, unless specified |
| $\cdot$ | Dot product |
| $x$ | Input of Range ratio equation |
| $*$ | Highest accuracy for all experimental results |

Table 1: Notation Table for Defined Symbols and Their Meanings in Our Study

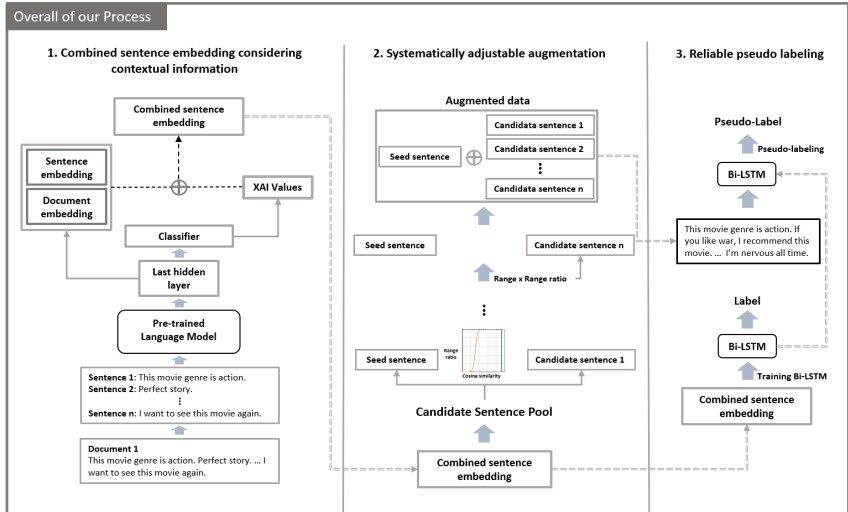

Figure 1: The proposed augmentation process in this study consists of three main steps: 1) Combined sentence embedding considering contextual information, 2) Systematically adjustable augmentation, and 3) Reliable pseudo-labeling.

# 3 METHODS

## 3.1 COMBINED SENTENCE EMBEDDING CONSIDERING CONTEXTUAL INFORMATION

As shown in Figure 1, the first step of the process involves combined sentence embedding that takes contextual information into account. This step consists of two components: 1. SHAP value extraction, and 2. Combining sentence embedding & conversion into sequence form. First, to generate the combined sentence embedding, a SHAP module is applied to the fine-tuned BERT's classification head to extract the Token SHAP Values. Second, using documents and sentences as inputs to BERT, document and sentence embeddings are extracted from the last hidden layer. In other words, the newly defined combined sentence embedding, which combines three extracted pieces of information, encompasses the original information of the sentence, including the emotion and semantic information conveyed in the original text.

### 3.1.1 EXTRACT TOKENS SHAP VALUES

As shown in the combined sentence embedding part of Figure 2, we used the fine-tuned BERT and token SHAP module to extract the SHAP values for each token present in the original document. Token SHAP values with red values indicate negatively influential tokens for the actual positive label, whereas blue values represent tokens that have a positive impact on the positive label. The extracted token SHAP values are used as materials for both document and sentence embedding, leading to the creation of combined sentence embedding that incorporates information from the original text, along with emotional and semantic information within the text.

### 3.1.2 COMBINED SENTENCE EMBEDDING & CONVERSION TO SEQUENCE FORM

The structure to generate the proposed combined sentence embedding considering contextual information and converting the generated combined sentence embedding into a sequence form, which is used as input for assigning pseudo-label, is illustrated in Figure 2.

**Combined sentence embedding** is the first structure; we gain sentences by using Sentence-tokenizer about documents. These documents and sentences were then used as inputs to the fine-tuned BERT model to extract the token Embeddings, which were the outputs of the last hidden layer. Additionally, we combined the extracted Token SHAP Values from the previous step, Extract SHAP Values, to create Token-SHAP Embeddings that incorporate emotional information.

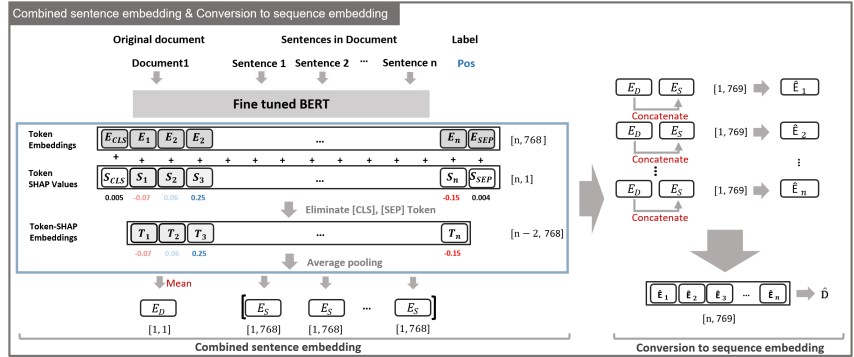

Figure 2: The structure for generating the proposed combined sentence embedding in considering contextual information and the structure for converting the generated combined sentence embedding into a sequence form, which is used as input for assigning Pseudo-labels.

Next, we combined the token SHAP values and token embeddings to generate Sentence-level embeddings, $E_S$, with a dimensionality of 768 through average pooling. For Document-level embeddings, we took the average, resulting in a 1-dimensional vector, $E_D$.

Consequently, we concatenate a 1-dimensional $E_D$ with a 768-dimensional $E_S$ to form $\hat{E}$, which has a dimensionality of 769. $\hat{E}$ contains information about the original text and the emotions and meanings associated with each sentence.

**Conversion to sequence form** is the second structure; the conversion to sequence form demonstrates the process of transforming the generated $\hat{E}$ into the input format suitable for a sequence model. We stacked the previously obtained $\hat{E}$ embeddings for each sentence to create a sequence vector that represents the document-level format. This transformation results in an embedding denoted by $\hat{D}$ with dimensions of [length, 769], which can be utilized as an input for a sequence model.

Moreover, we considered the characteristics of each dataset in this study and determined the average sentence count in a document as the sequence length. To ensure a fixed-length vector format, zero-padding and truncation were applied to the embeddings.

## 3.2 SYSTEMATICAL RANDOMNESS ASSIGN AUGMENTATION

As mentioned earlier, the second step of the process, systematical randomness, assigns augmentation and consists of two components: 1. Define candidate sentence pool, 2. Systematically adjust the candidate sentence pool range 3. Utilizing Systematically Adjustable Approaches for Data Augmentation.

### 3.2.1 DEFINE CANDIDATE SENTENCE POOL

The method used to define the candidate sentence pool required for augmentation is illustrated in Figure 3. As shown in Figure 3, we extracted the existing sentences and created a set of $\hat{E}$ for all the sentences obtained earlier, defining it as the Sentence Pool.

### 3.2.2 SYSTEMATICALLY ADJUST THE CANDIDATE SENTENCE POOL RANGE

The range of the candidate sentence pool was systematically adjusted by assigning randomness. The following describes the process of augmentation by systematically adjusting the randomness.

Let *Cs* represent the semantic cosine similarity between two sentences. Given embeddings *A* and *B* of two sentences, the cosine similarity *Cs* between *A* and *B* can be calculated as follows:

$$Cs(A, B) = \frac{A \cdot B}{||A||||B||} \tag{1}$$

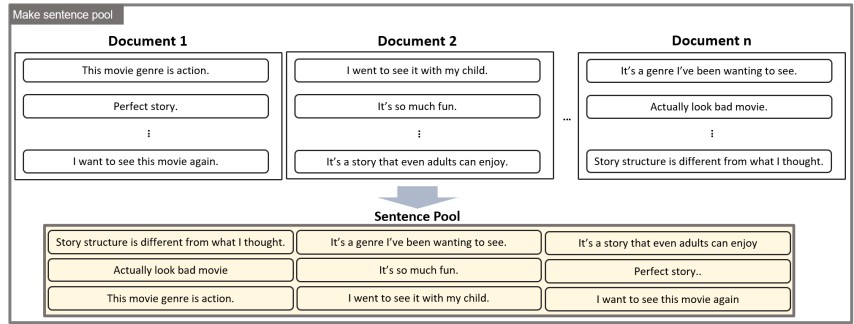

Figure 3: A method for creating a Candidate sentence pool.

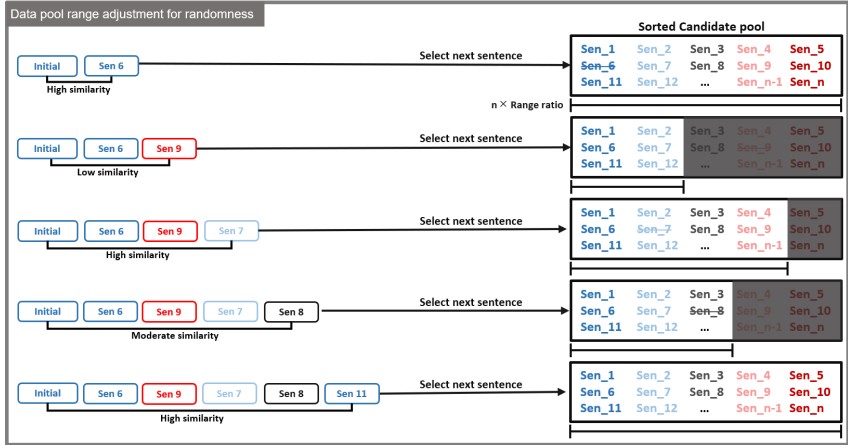

Figure 4: Augmentation method to adjust the level of manipulation by assign randomness.

Next, let's define *Seed(Ê)* as the $\hat{E}$ of the Seed sentence, *Random(Ê)* as the $\hat{E}$ of the randomly selected sentence in the candidate sentence pool. We can calculate the scaled input value *x* for the range ratio equation, which systematically assigns randomness using the following equation:

$$x = \frac{Cs(Seed(\hat{E}), Random(\hat{E})}{\displaystyle\sum_{i=1}^{n} Cs(Seed(\hat{E}), \hat{E}_i)} \tag{2}$$

Once we obtain the value of *x*, the range ratio for determining the range of the candidate sentence pool can be calculated as follows:

$$Range\ ratio = \frac{e^{10(\alpha x - m)}}{10 + e^{10(\alpha x - m)}} \tag{3}$$

In other words, the range ratio serves as a proportion for adjusting the range of the candidate sentence pool that can be selected, and the hyperparameters for the range ratio are denoted by $\alpha$ and *m*.

### 3.2.3 UTILIZING SYSTEMATICALLY ADJUSTABLE APPROACHES FOR DATA AUGMENTATION

Figure 4 illustrates the practical process of augmenting data by systematically adjusting the range of the candidate sentence pool to introduce randomness. This augmentation method allowed us to control the manipulation level. We begin by selecting a seed sentence from the candidate sentence pool and calculating the cosine similarity between the seed sentence and all other sentences. The Candidate sentence pool is sorted in descending order based on these cosine similarities. Next, we used the calculated *x* value as an input to the range ratio equation, which allowed us to systematically adjust the range and select the combined sentence embedding.

In Figure 4, when calculating *x* for the seed and second selected sentences, we obtained a high value, indicating high similarity. Consequently, we increase the range ratio to introduce high randomness.

This widening of the candidate sentence range increases the diversity of selectable sentences. Conversely, when the $x$ value for the seed sentence and the randomly selected sentence in the adjusting range is low, indicating low similarity, we decrease the range ratio to introduce low randomness. This narrowing of the candidate sentence range encourages the selection of $\hat{E}$ with a meaning similar to that of a seed. As a result, the augmented data $\hat{A}$ have a combined form of combined sentence embedding $\hat{E}$.

### 3.3 RELIABLE PSEUDO-LABELING

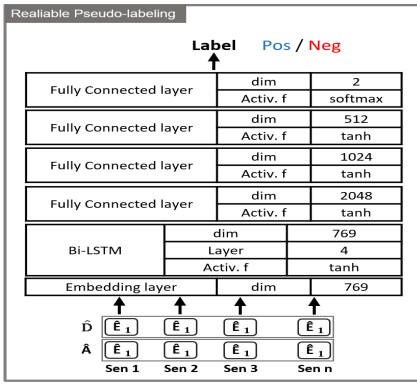

Figure 5: Structure of Bi-LSTM

In the final stage of our proposed method, we present the process of assigning reliable pseudo-labels to the augmented data $\hat{A}$. This process involves training the sequence form of $\hat{D}$, which is generated in the conversion to the sequence form step.

First, as shown in Figure 5, Bi-LSTM is trained using the sequence-based $\hat{D}$, which is the combined sentence embedding based on $\hat{E}$ for the original document in the conversion to the sequence form step. Next, the trained Bi-LSTM takes $\hat{A}$ generated from the systematical randomness assignment augmentation as an input to assign reliable pseudo-labels, which are then added to the training dataset. Bi-LSTM, trained on combined sentence embedding that incorporates various types of information, can assign more reliable labels to augmented data compared to conventional methods using semi-supervised learning.

## 4 EXPERIMENTS

### 4.1 EXPERIMENTAL SETUP

|  | BERT | | | Bi-LSTM | | |
|---|---|---|---|---|---|---|
|  | IMDB | SEM-Eval17 | YELP | IMDB | SEM-Eval17 | YELP |
| Batch size | 16 | 32 | 32 | 2 | 2 | 2 |
| Learning rate | 2e-5 | 2e-5 | 2e-5 | 1.5e-4 | 1.5e-4 | 1.5e-4 |
| Epoch | 5 | 5 | 5 | 10 | 10 | 10 |

Table 2: Hyper parameters of BERT & Bi-LSTM.

In the experiment, we compared the performance of existing text augmentation methods, namely Easy Data Augmentation(EDA), WordNet, NLP Albumentation, Back-Translation, Self-Supervised Manifold Based Data Augmentation(SSMBA), An Easier Data Augmentation(AEDA), and Pegasus Augmentation, using the proposed methodology. To ensure a fair comparison of the effectiveness of the proposed approach, we set up an experimental environment that was as close as possible. All experimental results are averaged of 5 runs, numbers are in percentages. Additionally, detailed information regarding an ablation study comparing the performance of the proposed systematic range ratio with various hyperparameters $\alpha$ and $m$ is available in Appendix A.

We used BERT-based(Devlin et al., 2019) from Hugging Face for data training and extracted token-wise importance using SHAP on the trained model. Furthermore, we used Bi-LSTM(Cornegruta et al., 2016) to assign reliable pseudo-labels to augmented data. The hyperparameter information for BERT and Bi-LSTM used in the experiments is summarized in Table 2.

### 4.2 DATA DESCRIPTION

To evaluate the performance of the proposed augmentation method, we conducted experiments on text classification tasks using three datasets with sentiment polarity classification capabilities: IMDB, SEM-Eval17, and YELP. The summary statistics for each dataset are listed in Table 3.

| Dataset | Num of classes | Train set | Test set | N |
|---|---|---|---|---|
| IMDB | 5 | 500 | 5725 | 20 |
| SEM-Eval17 | 3 | 500 | 6300 | 7 |
| YELP | 5 | 500 | 10000 | 10 |

Table 3: Information about IMDB, SEM-Eval17, YELP Dataset. N means count of average sentence in each dataset, which to define a input sequence length of Bi-LSTM.

Furthermore, for sentiment classification, we re-labeled the IMDB and YELP datasets, categorizing reviews with ratings of 2 or lower as negative and those with ratings of 3 or higher as positive. Similarly, for the SEM-Eval17 dataset, we conducted experiments by selecting positive and negative data, excluding neutral examples.

Assuming a low-resource scenario, we transformed a dataset comprising 500 random samples into a binary class for experimentation. The number of augmented data points was set to 500, 1000, 2000, 3000, 4000 and 5000. We extracted the average count of sentences from each dataset to train the Bi-LSTM to assign pseudo-labels to the augmented data. The experimental process involved augmenting the data using the proposed methodology and then retraining BERT by adding augmented pseudo-labeled data to the existing training dataset.

**IMDB** (Maas et al., 2011) contains movie reviews and is labeled with positive and negative.
**SEM-Eval17** (Rosenthal et al., 2017) comprises reviews for various restaurants. It includes reviews written with star ratings, encapsulating customers' experiences with restaurants or services.
**YELP** (Zhang et al., 2015) is curated for sentiment analysis tasks, aiming to classify positive, negative, and neutral sentiments in diverse sentences and documents.

## 4.3 EXPERIMENTAL RESULTS

| Dataset | IMDB / base : 88.23 | | | | | SEM-Eval17 / base : 87.92 | | | | | YELP / base : 94.24 | | | | |
|---|---|---|---|---|---|---|---|---|---|---|---|---|---|---|---|
| | 500 | 1000 | 2000 | 3000 | Avg | 500 | 1000 | 2000 | 3000 | Avg | 500 | 1000 | 2000 | 3000 | Avg |
| EDA | 89.59 | 88.72 | 89.59 | 89.33 | 89.31 | 85.72 | 89.08 | 89.26 | 87.77 | 87.96 | 94.60 | 94.00 | 94.34 | 94.36 | 94.33 |
| WordNet | 89.47 | 88.93 | 89.54 | 89.69 | 89.41 | 88.26 | 88.63 | 89.24 | 89.21 | 88.84 | 94.03 | 94.20 | 94.56 | 94.38 | 94.29 |
| Albumentation | 88.66 | 90.04 | 89.55 | 89.87 | 89.53 | 89.50 | 87.72 | 88.34 | 89.00 | 88.64 | 94.23 | 94.26 | 94.23 | 94.29 | 94.25 |
| Back-Translation | 89.64 | 89.52 | 89.76 | 90.01 | 89.73 | 88.26 | 88.63 | 89.24 | 89.40 | 88.88 | 94.38 | **94.65** | 94.55 | 94.46 | 94.51 |
| SSMBA | 89.45 | 89.89 | **90.27** | 89.87 | 89.87 | 87.32 | 86.66 | 88.61 | 89.89 | 88.12 | 93.23 | 94.10 | 90.58 | 91.36 | 92.32 |
| AEDA | 87.74 | 89.75 | 89.64 | 89.83 | 89.24 | 89.76 | 87.95 | 89.60 | 89.60 | 89.23 | 94.62 | 94.23 | 94.57 | 94.20 | 94.41 |
| Pegasus | **89.78** | 90.17 | 90.20 | 89.78 | 89.98 | 89.37 | **90.13** | 90.05 | 89.47 | 89.76 | 94.40 | 94.22 | 94.55 | 94.43 | 94.40 |
| Ours(BERT-base) | 89.68 | **90.41***| 90.17 | **90.20** | **90.12** | **90.02** | 90.00 | **90.92***| **90.02** | **90.24** | **94.89***| 94.60 | **94.58** | **94.56** | **94.66** |

Table 4: Comparing average performance of ours and baselines across all datasets on different the number of augmented data points. Scores are the average of 5 runs. Numbers are in percentages.

The performances of the proposed method and the baselines for the 500 training datasets are listed in Table 4. Our approach outperformed the baselines in terms of performance improvement on all the datasets evaluated, with an increase of 2.18% on the IMDB dataset, 3.0% on the SEM-Eval17 dataset, and 0.65% on the Yelp dataset. The results were obtained in an environment where the hyperparameters alpha and m of the Range Ratio equation were set to 5 and 4.4, respectively.

Tables 5, 6 and 7 present a tabulation of the maximum, minimum, and average performance enhancements achieved by a single model for both the baseline and our proposed method. By employing systematic randomness assigning, instances where the minimum performance is marginally lower compared to the baseline due to the inclusion of sentences chosen as Noise exist. However, it is noteworthy that our method demonstrates the highest magnitude of improvement in maximum performance compared to other baseline techniques, with gains of 2.18% on the IMDB Dataset.

## 5 CONCLUSION

In conclusion, we have addressed the major issue in text augmentation, which is the level of manipulation, by proposing a method that overcomes the limitations of previous approaches, which

| | | | | IMDB / base : 88.23 | | | | |
|---|---|---|---|---|---|---|---|---|
| | EDA | WordNet | Albument | Back-translation | SSMBA | AEDA | Pegasus | Ours |
| Max | 1.36 | 1.46 | 1.81 | 1.78 | 2.04 | 1.83 | 1.97 | **2.18** |
| Min | 0.49 | 0.70 | 0.43 | 1.29 | 1.22 | -0.49 | **1.55** | 1.45 |
| Avg | 1.08 | 1.18 | 1.30 | 1.50 | 1.64 | 1.01 | 1.75 | **1.89** |

Table 5: Comparing the Max, Min, and Avg Performance Gains of Our Method and Baselines on the IMDB Dataset

| | | | | SEM-Eval17 / base : 87.92 | | | | |
|---|---|---|---|---|---|---|---|---|
| | EDA | WordNet | Albument | Back-translation | SSMBA | AEDA | Pegasus | Ours |
| Max | 1.34 | 1.32 | 1.58 | 1.48 | 1.97 | 1.84 | 2.21 | **3.00** |
| Min | -2.20 | 0.34 | -0.20 | 0.34 | -0.60 | 0.03 | 1.45 | **2.98** |
| Avg | -0.13 | 0.83 | 1.02 | 1.31 | 0.49 | 1.35 | 1.74 | **2.08** |

Table 6: Comparing the Max, Min, and Avg Performance Gains of Our Method and Baselines on the SEM-Eval17 Dataset

| | | | | YELP / base : 94.24 | | | | |
|---|---|---|---|---|---|---|---|---|
| | EDA | WordNet | Albument | Back-translation | SSMBA | AEDA | Pegasus | Ours |
| Max | 0.36 | 0.32 | 0.05 | 0.41 | -0.14 | 0.38 | 0.31 | **0.65** |
| Min | -0.24 | -0.21 | -0.01 | 0.14 | -3.66 | -0.03 | -.02 | **0.32** |
| Avg | 0.09 | 0.05 | 0.01 | 0.27 | 0.08 | 0.17 | 0.16 | **0.42** |

Table 7: Comparing the Max, Min, and Avg Performance Gains of Our Method and Baselines on the YELP Dataset

either rely on original labels or stop at a simple weighted sum level. Our proposed method involves systematically adjustable text augmentation and the utilization of document embedding, sentence embedding, and XAI values to create Combined sentence embedding, enabling the assignment of reliable pseudo-labels.

We compared our proposed method with seven existing augmentation methods on three datasets and observed superior results. Our augmentation method allows for the appropriate manipulation setting based on the diverse characteristics of domain-specific data and enables the assignment of reliable pseudo-labels. Specifically, by systematically assigning randomness to the Candidate sentence pool range for augmentation and training models with Combined sentence embedding, we achieve robust and reliable pseudo-labels, contributing to improved model performance.

However, our proposed method has limitations. When augmenting datasets consisting of single sentences, augmentation based on systematic randomness assignment at the word or token level is required, and the Range ratio equation needs to be tailored to each dataset.

Future research directions include exploring the possibility of finding optimal systematic equations that yield better performance for each dataset's domain, as well as considering the use of unlabeled datasets with similar properties to apply data augmentation.

Nevertheless, our proposed method significantly contributes to model performance, and we anticipate that our research will help address the crucial issue of the level of manipulation in text augmentation in the field of natural language processing. To the best of our knowledge, our approach is the first augmentation study that systematically assigns randomness to adjust the level of manipulation.

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

## A  PERFORMANCE COMPARISON VARYING HYPER PARAMETER $\alpha$, M OF SYSTEMATICAL RANGE RATIO EQUATION

Our model uses a Systematically adjustable randomness assignment approach that regulates the level of manipulation. Therefore, the experiment was conducted by augmenting 1000 data for each dataset by changing the parameters $\alpha$ and $m$ for the proposed equation. If $\alpha \approx m$, the probability of selecting a sentence that is highly related to the seed sentence increases by selecting a sentence with a high probability value based on cosine similarity; as $m$ decreases, the probability of selecting a sentence with less relatedness increases. we conducted experiments to investigate the effect of coefficient $\alpha$ and $m$ on the performance. We tested the range of $\alpha$ {1, 2, 3, 4, 5} and $m$ {1, 2, 3, 4, 4.4}.

As shown in Table 8, 9 and 10 We achieved the best performance when $\alpha$ was 3 and m was set to 1 in IMDB dataset, when $\alpha$ was 1 and m was set to 1 in SEM-Eval17 dataset and when $\alpha$ was 5 and m was set to 1 in YELP dataset.

In other words, The performance of the model was highest when $\alpha > m$ in IMDB, YELP and $\alpha = m$ in SEM-Eval 17 Dataset. Furthermore, it can be observed that for IMDB, SEM-Eval17, and YELP, the actual performance improvements are 90.57%, 91.15%, and 94.95%, respectively. These values are slightly higher by 0.16%, 0.23%, and 0.06%, respectively, compared to the previously mentioned

| | m = 1 | m = 2 | m = 3 | m = 4 | m = 4.4 | Avg $\alpha$ |
|---|---|---|---|---|---|---|
| $\alpha = 1$ | 90.38 | - | - | - | - | 90.38 |
| $\alpha = 2$ | 89.82 | 89.96 | - | - | - | 89.89 |
| $\alpha = 3$ | 90.52 | **90.57**\* | 90.53 | - | - | 90.54 |
| $\alpha = 4$ | 90.18 | 89.87 | 89.99 | 89.61 | - | 89.84 |
| $\alpha = 5$ | 89.96 | 90.17 | 90.03 | 90.01 | 89.61 | 89.96 |
| Avg m | 90.17 | 90.07 | 90.18 | 89.81 | 89.61 | - |

Table 8: Ablation study result on IMDB.

| | m = 1 | m = 2 | m = 3 | m = 4 | m = 4.4 | Avg $\alpha$ |
|---|---|---|---|---|---|---|
| $\alpha = 1$ | **91.15**\* | - | - | - | - | 91.15 |
| $\alpha = 2$ | 89.45 | 89.43 | - | - | - | 89.44 |
| $\alpha = 3$ | 90.95 | 90.74 | 89.56 | - | - | 90.42 |
| $\alpha = 4$ | 88.69 | 90.79 | 90.16 | 88.74 | - | 89.60 |
| $\alpha = 5$ | 88.68 | 90.39 | 89.97 | 90.57 | 90.40 | 90.00 |
| Avg m | 89.78 | 90.34 | 89.90 | 89.66 | 90.40 | - |

Table 9: Ablation study result on SEM-Eval17.

| | m = 1 | m = 2 | m = 3 | m = 4 | m = 4.4 | Avg $\alpha$ |
|---|---|---|---|---|---|---|
| $\alpha = 1$ | 94.58 | - | - | - | - | 94.58 |
| $\alpha = 2$ | 94.36 | 94.36 | - | - | - | 94.36 |
| $\alpha = 3$ | 94.74 | 94.76 | 94.53 | - | - | 94.68 |
| $\alpha = 4$ | 94.32 | 94.37 | 94.38 | 94.42 | - | 94.37 |
| $\alpha = 5$ | **94.95**\* | 94.55 | 94.44 | 94.43 | 94.21 | 94.52 |
| Avg m | 94.59 | 94.51 | 94.45 | 94.43 | 94.21 | - |

Table 10: Ablation study result on YELP.

maximum performance of our method, which was 90.41% for IMDB, 90.82% for SEM-Eval17, and 94.89% for Yelp.

Consequentially, the method of generating a review text by adding noise is efficient when the document contains many sentences. Conversely, selecting sentences with high cosine similarity is more efficient when the document contains only a few sentences. In addition, because the performance changes according to the parameters $\alpha$ and $m$ appear for each dataset, it can be confirmed that optimal data augmentation can be performed by setting parameters suitable for the various characteristics of each dataset.

