# OpenReview forum: "Regulating the level of manipulation in text augmentation with systematic adjustment and advanced sentence-embedding"
_ICLR.cc/2024/Conference — Submitted to ICLR 2024_

### Official Review · Reviewer_xUBm · 2023-10-23

**Soundness:** 2 fair
**Presentation:** 2 fair
**Contribution:** 2 fair
**Rating:** 3
**Confidence:** 4

**Summary:**

The authors propose two means of performing better data augmentation for (some?) NLP tasks.

First, the authors propose a method to systematically adjust the degree of randomness during the augmentation process. This allows them to generate augmented data that is on the one hand diverse, but on the other hand not too different from the seed data, as data that is too different from the seed data may get annotated with incorrect pseudo-labels.

Second, the authors propose a method that uses sentence embeddings, document embeddings, and Shapley values to augment data in a way that allows the authors to reliably generate pseudo-labels for generated data, even if the generated data is quite different from the seed data.

**Strengths:**

- The paper discusses data augmentation for natural language processing. This is an area that many practitioners care about.
- Sections 2.2 and 2.2 situate the paper’s work nicely.

**Weaknesses:**

- The code included in the supplementary materials of this submission does not include code used to evaluate the baseline methods. How did the authors evaluate the baseline methods?
- There is mention of Hyperparameter tuning for the proposed method, but not for the baseline methods. Were the baseline methods hypertuned?
- The authors compare their model with the Pegasus method (https://arxiv.org/pdf/1912.08777.pdf), EDA method, and AEDA method, but use evaluation datasets that are entirely non-overlapping with the datasets that the Pegasus/EDA/AEDA authors evaluated on. This makes it harder to be sure that the Pegasus, EDA and AEDA methods were given a fair shot in the comparison, especially since no code or hyper parameters were published by the paper reviewed here.
- The paper is submitted under “Datasets & benchmarks”, but the paper doesn’t describe a new dataset or benchmark.
- The paper is hard to read. For example, Section 3.2.1 states that “[the] method used to define the sentence pool required for augmentation is illustrated in Figure 3.” But from the figure, the only thing that becomes apparent is that a set of documents is transformed into a sentence pool, without any details about how this happens.
- The paper’s scope is not clear. Which NLP tasks (e.g. text classification, text summarization) does the method described in the paper apply to?
- The paper contains numerous small grammatical errors, e.g. “diverse researched” should be “diverse research”.
- The scores in Table 5 vary widely between max and min values. The authors should use many more than 5 runs to reliably compare the performance of the models.
- The authors appear to use a BiLSTM for classification. Would a transformer model perform better?
- The authors describe two independent ideas (controlling level of randomness in augmentation, and a sentence-embedding methodology) in one paper. Have the authors considered separating these into two papers? This would make it easier to investigate the two ideas independently, from a scientific perspective.

**Questions:**

- It appears that the paper’s method is for text classification. If my guess is correct, could the authors mention this in the abstract and introduction? This would help the readers grasp the paper more easily.
- Section 3.1 begins with “As shown in Figure 1, the first step in the process involves …”. Here, it would be useful to spell out which process is being discussed, e.g. “augmentation process”. Of course the reader can go to the figure and check, but that disrupts the reading.
- Why is the Range Ratio (equation 3) needed? Why can we not threshold on x directly, instead of first transforming x using a monotonic function, and thresholding afterwards?
- The fonts in the figures are too small to be legible when the paper is printed.
- The abbreviation “XAI” should be spelled out in the abstract.
- In the abstract, “incorporates” should be “incorporate”
- The paper mentions “utilized a logistic regression-based equation”. This is confusing, as the connection to logistic regression is very loose, and no model is being trained in this context.
- The third paragraph of Section 2.1 is almost verbatim the second paragraph of Section 1. This should be revised to avoid needless repetition.
- The authors typeset citations by writing \cite immediately after the word preceding it, which results in text like “CAM(Zhou et al., 2016)”. Could the authors please instead use ~\cite (with a tilde before the \cite), so that the text gets typeset like “CAM (Zhou et al., 2016)”  (with a space after “CAM”)?
- The authors write “calculates the magnitude of errors through back-propagation with respect to the input of the model”, but it should be “the gradient of errors”
- The paper states “… BBM tends to have less accurate interpretability than the ABM because of its fast computational ability”. Did the authors mean this, and if so, could they please explain it?
- In Table 1, the authors write “Euclidean, unless otherwise specified”. I could not find an alternative meaning of the norm specified anywhere in the paper. Can “unless otherwise specified” be removed?
- In Table 1, can “Typical meaning” be replaced with “Meaning”?
- Something is broken in the text snippet “conditional mean, computes the conditional mean”  in Section 2.2
- Something is broken in the text snippet “In this study utilize the” in Section 2.2
- Many more grammatical errors or broken sentences. Please review the paper for grammar mistakes one more time.
- Section 3.1 mentions “the fine-tuned BERT’s classification head”. This felt abrupt. Could the authors introduce the fine-tuned model appropriately?

---

### Official Review · Reviewer_HBAq · 2023-10-29

**Soundness:** 2 fair
**Presentation:** 1 poor
**Contribution:** 2 fair
**Rating:** 1
**Confidence:** 4

**Summary:**

This paper introduces a new text augmentation technique which is able to control the diversity of augmentations. They utilize information from sentence embedding and document embedding in this approach. Experimental results on three text classification datasets show this method outperforms the baselines.

**Strengths:**

1. This work has provided codes in supplementary materials to give us a reference to check the reproducibility.
2. Experimental results on three text classification benchmark datasets show the proposed method is effective.

**Weaknesses:**

1. The writing and presentation of this paper are poor, so this paper is hard to follow.

    Firstly, some sentences are not clearly organized and the logic is not good. For example, some sentences are really hard for readers to understand: "*The level of manipulation is the most important issue in text augmentation; low-level manipulation generates data similar to the original, resulting in inefficient augmentation because it cannot ensure diversity, whereas high-level manipulation causes reliability issues for labels and degrades the model’s performance. Therefore, ...*".

    Secondly, there are many typos such as *It **introduced** diversity into the training process* -> *It **introduces** diversity into the training process*.

    Thirdly, some abbreviations should be explained before being used. For example, this paper does not explain what is *XAI* in abstract.

2. The motivation is not clear. This paper does not explain why the proposed method address *the level of manipulation* issue. Some previous work like AutoAugment[1] also presented techniques to automatically search for improved data augmentation policies.

3. This paper omits many important references such as AutoAugment[1]. Also, it does not compare with these methods.

4. The improvements of performance are marginal, where the average improvements over the state-of-the-arts are all smaller than 0.5 in Table 4. Thus, we are not sure whether this approach is really effective and robust.

[1] Cubuk et al., AutoAugment: Learning Augmentation Policies from Data. CVPR 2019.

**Questions:**

See Weaknesses above. We encourage the authors to carefully modify and improve this paper.

---

### Official Review · Reviewer_WQ8f · 2023-11-01

**Soundness:** 2 fair
**Presentation:** 2 fair
**Contribution:** 3 good
**Rating:** 5
**Confidence:** 4

**Summary:**

The paper presents a novel text augmentation technique aiming to strike a balance in the level of manipulation applied to text data, addressing a common challenge in NLP. It introduces a method to systematically adjust the data manipulation pool and proposes a sentence-embedding methodology for robust pseudo-labeling. The authors validate the approach through extensive experiments, demonstrating improved performance compared to existing methods.

**Strengths:**

1) The paper introduces a methodical and innovative approach to text augmentation, employing a logistic regression-based equation and cosine similarity to dynamically adjust the candidate sentence pool. This ensures a tailored balance between diversity and semantic similarity, directly addressing and systematically controlling the critical issue of manipulation level in text augmentation.
2) The integration of sentence, document embedding, and XAI information for pseudo-labeling is innovative

**Weaknesses:**

I hope my reviews will help make the work more robust.

1) The authors highlight the challenges associated with preserving label reliability amid variations in text due to manipulation, with a particular focus on concerns about the reliability of newly assigned pseudo-labels. However, there are more advanced data augmentation methods available, such as those described in references [1], [2], and [3]. Reference [1] introduces a language model specifically trained to generate augmentations tailored to the class label, ensuring consistency between the input class label and the output augmentation. Reference [2] employs a multi-level optimization framework to tailor the augmentation process, benefiting the downstream classification model and maintaining class label consistency via a feedback mechanism. Reference [3] suggests a soft-augmentation approach based on embedding space, synthesizing gradients of all unseen words using a task-dependent similarity matrix. A comparison with these methods **or** a discussion of the proposed method's advantages over them would provide valuable insights. I understand it is challenging to compare with all these methods in limited time frame. However, a discussion would be insightful.
2) The impact of the SHAP module, as well as the utilization of document and sentence embeddings remains unclear from an empirical standpoint. Including an ablation study in the paper, which breaks down and analyzes the role of each component, would help clarify their contributions and improve understanding.
3) The experiments in the paper utilize BERT-base to demonstrate the effectiveness of the proposed text augmentation technique. To strengthen the evaluation, it would be insightful to extend the comparisons to include less powerful models, such as LSTM. An examination across a range of model complexities would offer a more robust validation of the proposed method.
4) More details about SHAP in the paper would make it easier for readers to better understand it.
5) It would be insightful to include the generated augmentations. I have a follow up question on it. (below)
6) It would be insightful to have the standard deviation of the method displayed.

[1] Anaby-Tavor, Ateret, et al. "Do not have enough data? Deep learning to the rescue!." Proceedings of the AAAI Conference on Artificial Intelligence. Vol. 34. No. 05. 2020.

[2] Somayajula, Sai Ashish, Linfeng Song, and Pengtao Xie. "A multi-level optimization framework for end-to-end text augmentation." Transactions of the Association for Computational Linguistics 10 (2022): 343-358.

[3] Somayajula, S. A., Jin, L., Song, L., Mi, H., & Yu, D. (2023, July). Bi-level Finetuning with Task-dependent Similarity Structure for Low-resource Training. In Findings of the Association for Computational Linguistics: ACL 2023 (pp. 8569-8588).

**Questions:**

I hope my questions will add more insights to the paper.

1) Could you please elucidate the specific advantages brought by the inclusion of document embeddings, especially considering that sentence embeddings are already incorporated into the model?

2) I would appreciate some clarification on how the proposed system assigns pseudo labels, particularly in challenging scenarios where sentences contain contrasting sentiments or uncorrelated events with conflicting emotions. For example, in sentences like “Movie is good. The actors and director did a bad job” or “Movie is good. The dinner in Hollywood was bad”, the sentiments are mixed.  Could the authors provide examples of how the system handles such cases, and discuss the impact on pseudo label assignment? I understand that the use of cosine similarity and the systematic adjustment of the candidate pool aims to mitigate these issues. However, there might be instances where stochasticity in sentence selection could still lead to pseudo label inconsistencies. For instance, in a movie review task, the sentence “Movie is good. The dinner in Hollywood was bad” could yield a high cosine similarity due to the presence of terms like ‘Hollywood’, but the Bi-LSTM might assign a negative sentiment based on the word ‘bad’, leading to a potential discrepancy.  Understanding how the system navigates these complexities would provide valuable insights into its robustness and reliability in pseudo label assignment.

3) The range ratio, indicative of the manipulation level through its pool length, is calculated using cosine similarity, denoted as 'x.' Specifically, 'x' is determined by comparing the embedding 'Eˆ' of the seed sentence (Seed(Eˆ)) with the embedding of a randomly selected sentence from the candidate pool (Random(Eˆ)). This process systematically assigns a level of randomness for the range ratio calculation. However, it raises a question: Why is the range ratio dependent solely on a metric derived from a single randomly chosen candidate, rather than being based on an average or aggregate measure across the entire pool? Or alternatively there can be clustering based on the estimated x's for all the sentences in the pool and choose to have sentences from most related clusters.

4) Can you provide more details on how the SHAP module is contributing to our understanding or interpretation of the model's predictions? Specifically, I am interested in understanding if and how it can shed light on the impact or effectiveness of data augmentations. Are there any specific insights or trends that can be derived from the SHAP values regarding how different augmentations might be influencing the model’s predictions? It would be beneficial to discuss examples or specific cases to illustrate these points.

---

> ### Comment · Reviewer_WQ8f · 2023-11-21
>
> Since my questions are not adequately answered, I am reducing the score from 6->5.

---

### Official Review · Reviewer_MCTc · 2023-11-01

**Soundness:** 2 fair
**Presentation:** 1 poor
**Contribution:** 2 fair
**Rating:** 3
**Confidence:** 4

**Summary:**

The paper proposes a technique for data augmentation in text. The main goal is to provide a way to control diversity and fidelity (meaning preservation) to construct augmentation dataset. This is achieved using reassignment of similarity scores to incorporate a wide informative pool of candidate sequences. Based on the formulation, when the new similarity score is high, a wide pool is selected while a smaller pool of candidate sequence is selected in case of lower scores. Additionally different embeddings with Bi-LSTM is used to form supervised datapoints.

The resultant augmented data is used for training BERT models, and shows competitive results against comparable baselines.

That being said, this paper needs a lot of rework and would not recommend acceptance in its current form.

**Strengths:**

1. Comparison against multiple competitive baselines and better than baseline results using this method.

2. Combination of multiple source of information for generating pseudo labels for the training data.

**Weaknesses:**

1. The writeup of this paper needs to be improved. A lot of details are missing. Kindly see Question/Comments section for the same.

2. Qualitative Analysis missing. Why are the labels more reliable?

3. Statistical Significance testing missing for quantitative evaluation.

**Questions:**

1. Section 2.2: SHapley Additive exPlanation

2. Figure 2: It is difficult to understand this figure. How are the embeddings calculated. Where do the final tokens appear from? Kindly enumerate the equations governing the embedding construction.

3. Section 3.1: What is BERT fine-tuned on?

4. Section 3.1.2: Why is 1-D used for representing document embedding? What information does it contain? Kindly perform an ablation study to show that those document embeddings play a significant role in the embedding construction.

5. Section 3.2.3: Excessive use of the word systematically.

6. Section 3.3: Why is a Bi-LSTM used here instead of a transformer model? Why does it lead to reliable labels?

7. Section 4.3: Are the results mentioned statistically significant? Kindly report numbers from a statistical significance testing.

8. Section 4.3: 500 training datasets -> 500 training datapoints

9. Section 4.3: Ablation study of components missing.

10. Conclusion: Kindly mention quantitative metrics in the summary instead of re-writing the abstract here.

11. In the writeup, there are multiple instances where words have been titlecased. Kindly rectify those.

---

### Meta-Review · Area_Chair_RufU · 2023-12-05

**Metareview:**

This paper proposes a novel approach to text augmentation that is designed to balance the diversity and semantic fidelity of augmented text. The authors leverage a logistic regression-based formulation alongside cosine similarity to dynamically adjust the candidate sentence pool for augmentation, an idea that purports to deal with the challenge of manipulation levels in text augmentation pertinent to maintaining label reliability.

Strength:
The approach takes on the important issue of managing manipulation levels during text augmentation, aiming to provide diverse yet semantically faithful augmented data for model training.

Weakness:
However, all reviewers have consistent concerns about the presentation, baseline comparison and the experimental results of this paper. For example, the presentation of the paper struggles with clarity, both grammatically and structurally, leading to difficulty in understanding the methods and contributions. There is a lack of depth in the comparison with state-of-the-art methods, and  empirical results do not convincingly demonstrate the impact of various proposed components. Unfortunately, authors does not give response to the reviewers during the rebuttal period.

**Justification For Why Not Higher Score:**

All reviewers have consistent concerns about the presentation, baseline comparison and the experimental results of this paper.  Authors does not give response to the reviewers during the rebuttal period.  Ths manuscript should be significantly improved to fix these common concerns and now it is below the bordeline of acceptance.

**Justification For Why Not Lower Score:**

N/A

---

### Decision · Program_Chairs · 2024-01-16

Reject